# Micronuclei Detection by Flow Cytometry as a High-Throughput Approach for the Genotoxicity Testing of Nanomaterials

**DOI:** 10.3390/nano9121677

**Published:** 2019-11-24

**Authors:** Alba García-Rodríguez, Liliya Kazantseva, Laura Vila, Laura Rubio, Antonia Velázquez, María José Ramírez, Ricard Marcos, Alba Hernández

**Affiliations:** 1Department of Genetics and Microbiology, Faculty of Biosciences, Universitat Autònoma de Barcelona, 08193 Bellaterra, Spain; albagr.garcia@gmail.com (A.G.-R.); liliyak1292@gmail.com (L.K.); lauravilavecilla@hotmail.com (L.V.); laurarubio33@hotmail.com (L.R.); Antonia.Velazquez@uab.cat (A.V.); 2Consortium for Biomedical Research in Epidemiology and Public Health (CIBERESP), Carlos III Institute of Health, 28029 Madrid, Spain; 3Consortium for Biomedical Research on Rare Diseases (CIBERER), Carlos III Institute of Health, 28029 Madrid, Spain; MariaJose.Ramirez@uab.cat

**Keywords:** BEAS-2B cells, flow cytometry MN (FCMN) assay, TiO_2_NPs, ZnONPs, CeO_2_NPs, AgNPs, MWCNTs

## Abstract

Thousands of nanomaterials (NMs)-containing products are currently under development or incorporated in the consumer market, despite our very limited understanding of their genotoxic potential. Taking into account that the toxicity and genotoxicity of NMs strongly depend on their physicochemical characteristics, many variables must be considered in the safety evaluation of each given NM. In this scenario, the challenge is to establish high-throughput methodologies able to generate rapid and robust genotoxicity data that can be used to critically assess and/or predict the biological effects associated with those NMs being under development or already present in the market. In this study, we have evaluated the advantages of using a flow cytometry-based approach testing micronucleus (MNs) induction (FCMN assay). In the frame of the EU NANoREG project, we have tested six different NMs—namely NM100 and NM101 (TiO_2_NPs), NM110 (ZnONPs), NM212 (CeO_2_NPs), NM300K (AgNPs) and NM401 (multi-walled carbon nanotubes (MWCNTs)). The obtained results confirm the ability of AgNPs and MWCNTs to induce MN in the human bronchial epithelial BEAS-2B cell line, whereas the other tested NMs retrieved non-significant increases in the MN frequency. Based on the alignment of the results with the data reported in the literature and the performance of the FCMN assay, we strongly recommend this assay as a reference method to systematically evaluate the potential genotoxicity of NMs.

## 1. Introduction

Thousands of products containing nanomaterials (NMs) have been settled in the European industry and in regular consumer products over the last years, due to the new and useful properties of materials at the nanometric scale [1]. Several estimates suggest that the use of NMs will continue experiencing an exponential increase, as substantial economic and technical resources are being dedicated to the study and design of new NMs with industrial applicability [2]. Some of these NMs may be introduced into the environment or come into contact with humans, resulting in unexpected biological effects. Indeed, the effects of occupationally and environmentally exposed populations are uncertain. Thus, new information regarding their potentially harmful effects is urgently required.

It is known that the toxicity of NMs depends on their physicochemical characteristics, such as size, shape, surface area, chemical composition, surface charge, agglomeration/aggregation, and/or crystal structure, among others [3,4,5]. We are then facing a complex scenario where a huge amount of NMs needs to be tested under multiple modulatory variables. For this reason, high-throughput methods are required to generate rapid and robust data that can be used to assess and/or predict the potentially harmful effects of NMs in biological systems, as postulated by different authors [6,7,8]. Previous to the use of any analytical method, dispersion procedures are indispensable to get NMs uniform samples. It has been largely demonstrated that the use of different dispersion protocols can lead to variability in their suspensions and, consequently, to different physicochemical properties [9,10,11]. For this reason, we aimed to follow a protocol generated under the frame of the EU project NanoGenotox, a well-standardized protocol tested and already harmonized among laboratories. However, since this is a time-consuming process, the dispersion step may act as a real bottleneck preventing the development of emerging high-throughput methodologies to test the genotoxic properties of NMs. To overcome this problem, we recently proposed to freeze the NMs immediately after dispersion, as an alternative to working with freshly dispersed samples, since their physicochemical and biological characteristics were unaltered after the freezing procedure [5].

Among the different toxic effects that NMs are able to exert, DNA damage requires special attention. In previous studies, we clearly demonstrated (by using transmission electron microscope (TEM) images) the ability of BEAS-2B to uptake, mainly by endocytosis, big agglomerations of TiO_2_NPs, ZnONPs, CeO_2_NPs and multi-walled carbon nanotubes (MWCNT), after exposures lasting for 24 h [5]. Similar results were obtained when BEAS-2B cells were exposed to AgNPs for 24 h [12]. This phenomenon of internalization would act as the first step of a potential NPs-cell nucleus interaction and, consequently, to DNA impairment. Therefore, Nanogenotoxicology is emerging as a specific research field covering such demands [13,14]. Among the different methodological approaches aiming to detect and quantify DNA damage, the micronucleus assay is very popular as it is well-established, reliable, accurate and reproducible. This assay measures the incidence of micronuclei (MN), which are small cytoplasmic chromatin bodies resulting from the condensation of acentric chromosome fragments or whole chromosomes lagging during cell division. The relevance of MN estimations relies on the fact that they act as very sensitive indicators of genetic damage and as a good surrogate biomarker of cancer risk [15]. The classical MN assay use cytochalasin-B to arrest cytokinesis (CBMN assay) and MN are microscopically scored in arrested/binucleated cells [16]. Since the classical in vitro approach is labor-intensive, as it requires many hours of microscopic analysis, an alternative scoring of the MN frequency using flow cytometry (FC) was earlier proposed [17]. The proposed flow cytometry micronucleus (FCMN) assay involves the lysis of outer membranes previous to the use of one or more nucleic acid dyes to discriminate the released nuclei and MNs, according to the DNA-dye associated fluorescence intensities. An updated revision of the original method discussing advantages, limitations and future recommendations have been carried out [18]. The FCMN assay has already been used to test several NMs such as TiO_2_NPs [19,20], AgNPs [20,21,22,23] and SiO_2_NPs [24].

To identify what environmental health and safety aspects of nanomaterials are relevant from a regulatory point of view, the NANoREG EU project was launched. One of the aims was to determine the suitability of different in vitro assays in terms of reliability and predictive value. In such context we assessed the ability of six different NMs, namely NM100 and NM101 (TiO_2_NPs), NM110 (ZnONPs), NM212 (CeO_2_NPs), NM300K (AgNPs) and NM401 (MWCNT, multiwalled carbon nanotubes) of inducing MN, as detected by the FCMN assay, by using human bronchial epithelial cells (BEAS-2B). This cell line was selected because the lungs are considered the main target of NM exposure. Here we present the obtained results. The main objective of this study was to show that the FCMN assay is at least as sensitive to the CBMN assay, as developed by other partners of the consortium, in the detection of the genotoxic potential on NMs. In addition, the combined use of FCMN and frozen dispersions can permit its application in high throughput approaches.

## 2. Material and Methods

### 2.1. Selected Nanomaterials

The selected NMs were—TiO_2_NPs (NM100 and NM101), ZnONPs (NM110), CeO_2_NPs (NM212), AgNPs (NM300K) and MWCNT (NM401). All six NMs were supplied by the NANoREG consortium, who bought the materials and sent aliquots to all the participants. Although NMs were well characterized in the frame of the project [25], we determined nanoparticle size and morphology using a JEOL 1400 instrument (Jeol LTD, Tokyo, Japan). TEM sizes were calculated in dry conditions by measuring over 100 particles in random fields of view. Further characterization using dynamic light scattering (DLS) and dispersion in the whole cell culture medium was carried out. DLS analysis was performed on a Malvern Zetasizer Nano-ZS zen3600 (Malvern Panalytical, Malvern, UK) instrument. Nanoparticles were prepared and sonicated as previously described [26]. Just after sonication, aliquots were frozen in liquid nitrogen and stored at −80 °C. In this way, all the used concentrations of the different NMs were ready to be used just after thawing [5].

### 2.2. Cell Culture

The BEAS-2B cell line was chosen in the present study as a model system to test the usefulness of the FCMN technique, mainly due to the vast informative background existing in the nanotoxicology field, as provided by the literature. The BEAS-2B cell line was kindly provided by Dr. H. Karlson, from the Swedish Carolinska Institute. This is a bronchial cell line with epithelial morphology. Cells were cultured as a monolayer in 75 cm^2^ culture flasks coated with 0.03% collagen in serum-free bronchial epithelial cell growth medium (BEGM, Lonza, CA, USA). Log-phase cells were grown on 12 well plates without the 0.03% collagen coating, to determine, cytotoxicity and micronucleus frequency. Cells were maintained in a humidified atmosphere of 5% CO_2_ and 95% air at 37 °C.

### 2.3. Cell Viability

BEAS-2B cells were exposed to different concentrations of the selected NMs, according to previous toxicity data. Thus, the initial selected concentrations were as follows—NM100 (TiO_2_NPs, 10, 70, 140, 200 and 250 μg/mL), NM101 (TiO_2_NPs, 10, 70, 140, 200 and 250 μg/mL), NM110 (ZnONPs, 1, 3, 5, 7 and 10 μg/mL), NM212 (CeO_2_NPs, 10, 50, 80, 120 and 180 μg/mL), NM300K (AgNPs, 1, 3, 5, 7 and 10 μg/mL) and NM401 (MWCNT, 5, 10, 20, 30 and 50 μg/mL). These concentrations were tested to determine cell viability and the appropriate subtoxic concentrations to be used in the FCMN assay. The same procedure was done for the positive control (mitomycin C, MMC) and the tested concentrations were 0.010, 0.050, 0.075, 0.100 and 0.150 μg/mL. A negative control, represented by untreated cells with a cell culture medium, was also included. 

After treatments lasting for 48 h, cells were washed three times with 0.5 mL of phosphate buffer solution (PBS) (1%), incubated 3 min at 37 °C with 0.25 mL of trypsin (1.5%) to detach and individualize them. After that, cells were diluted (1/10) in ISOTON^®^, washed trice to eliminate dead cells and counted with a Z^TM^ Series coulter-counter (Beckman Coulter Inc., Brea, CA, USA) [5]. ISOTON^®^ is an isotonic solution used to keep BEAS-2B cells in stable condition while the device counts the cells.

Viability values for each concentration were calculated by averaging two independent viability experiments, each containing three replicates per sample.

### 2.4. The FCMN Assay

According to the cytotoxicity assay, five concentrations of each one of the six selected NMs were used. The criteria followed to choose the concentrations were deducted from the study of Bryce et al. (2007), where treatments exhibiting mean relative survival ≥ 40% were carried forward for further processing [27]. Cells were plated (120,000 per well) in 12 well plates and incubated for 48 h with the selected concentrations. This exposure time was an agreement inside the consortium. Every concentration and plate was done in duplicate. Negative control was represented by untreated cells with the cell culture medium (BEGM). A positive control (MMC, 10 µg/mL) was also included. Two different independent experiments were carried out. The FCMN assay was carried out as previously described [28] but introducing several modifications. Briefly, after the exposure time, the culture medium with the different concentrations of NMs was removed from the wells by aspiration. Cells were rinsed twice with phosphate-buffered saline (PBS) 1X, at room temperature and collected by trypsinization (trypsin 0.5X in PBS 1X). To inactivate the trypsin, PBS 1X with 2% of heat-inactivated fetal bovine serum was added to each well. Cells were resuspended in each well and later transferred (1 mL) to FC tubes, which were centrifuged at 1000 rpm for 8 min. After centrifugation the supernatant was aspired, leaving 50 μL. Cells were gently resuspended by tapping the FC tubes. 

After that, 125 μL of nucleic acid dye [0.125 mg/mL ethidium monoazide bromide (EMA) (25 μL) prepared in PBS 1X with 2% of heat-inactivated fetal bovine serum (100 μL)] were added to each tube, placed in a rack. EMA staining acts as a detector of early apoptosis. Samples were submerged 2 cm deep in the ice. A light source (60 W light bulb) was applied 30 cm above the tubes for 20 min. After that, 3 mL of cold PBS 1X with 2% of heat-inactivated fetal bovine serum was added to each sample and the tubes were protected from light by covering them with foil. Next, cells were centrifuged at 1000 rpm for 8 min and the supernatant was aspired leaving some 50 μL, to increase cell concentration. 

Cells were gently resuspended by tapping and maintained at room temperature for 20 min. After that, 250 μL of lysis-solution-1 (0.584 mg/mL of NaCl, 1 mg/mL of sodium citrate, 0.3 μL/mL IGEPAL^®^ (detergent), 1 mg/mL RNase A and 0.2 μM of SYTOX^®^ Green nucleic acid stain prepared in deionized water) were slowly added to each sample. The tubes were briefly vortexed at medium speed and incubated at room temperature for 1 h. Finally, 250 μL of lysis-solution-2 (85.6 mg/mL of sucrose, 15 mg/mL of citric acid and 0.2 μM of SYTOX^®^ Green prepared in deionized water) was added. Samples were kept at room temperature for 30 min, before the FC analysis. A FACSCanto (Becton Dickinson, Franklin Lakes, NJ, USA) cytometer was used. The fluorescent emission associated with SYTOX^®^ was collected in the isothiocyanate fluorescein channel (530/30). The fluorescent emission associated with EMA was collected in the PerCP-Cy5.5 channel (530/30). For each concentration, a total of 20,000 cells were scored and the number of MN was recorded. The frequency of MN was calculated by dividing the number of events placed on the “MN region” by those from the “nucleus region.” This number is multiplied by 1,000 and the values are expressed as MN per 1000 cells.

For a more easy understanding of the FCMN protocol, we have included a more detailed description in the Appendix A.

### 2.5. Statistical Analysis

At least two independent experiments, using triplicates of each treatment, were performed in the viability assays (*n =* 6). For the FCMN assay, three independent experiments using triplicates were carried out, reaching and *n* value of 9. The results were evaluated with FlowJo Ver. 10.0 (LLC, Ashland, OR, USA). One-way ANOVA with Tukey’s post-test was used to compare the differences between means. Data were analyzed with GraphPad Prism version 5.00 for Windows (GraphPad Software, San Diego, California, USA, http://graphpad.com). Differences between means were considered significant at *p* < 0.05.

## 3. Results and Discussion

### 3.1. Nanoparticles Characterization

In nanotoxicological studies, it is important to have strict control of the dispersion procedure when NMs are administered to cultured cells [29]. In our case, we have used the dispersion protocol generated in the frame of the NanoGenotox EU project [30] to ensure good NMs dispersions. Figure 1 shows representative TEM images of the used TiO_2_NPs, ZnONPs, CeO_2_NPs, AgNPs and MWCNTs. The mean sizes of NPs obtained from TEM images are shown in Table 1. They represent the average diameter with the exception of MWCNTs NM401, where we indicate the average length. Because NMs are administered in dispersion, the characteristics of the selected nanomaterials in the culture medium were analyzed by DLS. As indicated in Table 1, the average diameters measured by DLS were slightly larger than those measured from TEM images. This would indicate that some degree of aggregation takes place in dispersions. Due to the fibrillary characteristics of MWCNT NM401, measuring diameter by DLS is not appropriated for this type of material, because this method is only useful for those NPs with spherical-like forms. To evaluate the effects of time on NPs dispersion, in a previous study DLS measurements were carried 24 h after dispersion. Results did not seem to indicate that the incubation time has a relevant effect on the NMs agglomeration [31].

### 3.2. Cell Viability 

Prior to any type of genotoxic characterization of the selected nanomaterials, it is essential to define the range of concentrations to work with. To select such range, toxicity studies are required. Although there is a wide set of assays that can be used to measure toxicity, most of them show interferences when applied to nanomaterials, as reviewed for AgNPs [32]. Those viability assays using fluorescent dyes produced the highest interference, while those such as cell counter and flow cytometry methods were less prone to produce interferences. Using a cell colter-counter approach we have established an order of cytotoxicity for the selected nanomaterials (Figure 2). This is as follows—ZnONPs > AgNPs > TiO_2_NPs > CeO_2_NPs ≈ MWCNT; ZnONPs being highly toxic and non-relevant toxicity was detected for CeO_2_NPs and MWCNT, according to the used concentrations.

### 3.3. Genotoxicity

To see how the FCMN assay works in BEAS-2B cells we carried out a previous study by checking the spontaneous MN levels (negative control), as well as those induced by the reference mutagen mitomycin-C (positive control). As observed in Figure 3, the obtained results agree with the expectations. After that, the FCMN assay was carried out with the six selected NMs and the obtained results are indicated in Figure 4. No significant differences were detected in the MN frequency formation when the BEAS-2B were exposed either to both TiO_2_NPs NM100 and NM101 (Figure 4A,B), as well as when CeO_2_NPs NM212 were tested (Figure 4D). Nevertheless, an increase in the MN frequency was observed, exposing the cells to the highest dose (5 µg/mL) of ZnONPs, such an increase did not attain statistical significance (Figure 4C). However, AgNPs (7 µg/mL) and MWCNT (20 and 50 µg/mL) were able to increase the MN formation in a statistically significant way (Figure 4E,F). In all the cases we used the same concentration for the positive control.

Titanium dioxide nanoparticles have been largely evaluated from the genotoxic point of view but a clear conclusion has not been reached yet, since the in vitro and in vivo studies evaluating the genotoxicity of TiO_2_NPs present conflicting results [33]. Despite a large number of genotoxicity studies carried out with this nanomaterial, only a few of them used MN as a potential indicator of genotoxicity. From them, only two have recently used FC methodology to evaluate MN induction [19,20]. The evaluation of TiO_2_NPs in two different cell lines (BEAS-2B and HepG2) showed that positive effects were only observed when 10% of fetal bovine serum was used in the culture medium. The authors pointed out the relevant role of culture media and conclude that MN was only detected when the total amount of protein in the culture medium exceeds 1% [34,35]. According to these authors, the increased amount of proteins would act as avoiding NM agglomeration and permitting cell uptake. Nevertheless, high uptake was demonstrated in BEAS-2B cells but without the induction of genotoxic damage, as evaluated by the comet and the CBMN assays [26].

In a recent study, TiO_2_NPs with different crystalline phases (anatase, rutile and rutile/anatase mixture) were tested in human peripheral blood mononuclear cells and negative effects were observed in the CBMN assay, although oxidative DNA damage induction was detected [36]. These negative results agree with those reported by two different laboratories where male and female rats were exposed to six pigment-grade or ultrafine (anatase and/or rutile) TiO_2_NPs compounds. The authors concluded that the lack of MN induction was mainly due to the poor uptake through the gastrointestinal barrier [37]. Nevertheless, we have recently reported that TiO_2_NPs easily cross the in vitro model of the intestinal barrier constituted by differentiated Caco-2 cells. This cross did not disturb the integrity/functionality of the barrier and did not induce DNA damage in the differentiated Caco-2 cells [38].

From the two studies found in the literature evaluating MN induction by flow cytometry [19,20], only one used BEAS-2B cells and three different TiO_2_NPs (NM100, NM101 and NM103) at concentrations ranging from 1 to 30 µg/mL, two of them (NM100 and NM101) also used in our study. No MN induction was detected when NM100 and NM101 were tested in the CBMN assay, although a slight effect was observed for NM101 in one of the tested concentrations when FC was used, supporting the advantage of the MN scoring by the last methodology (results indicated in Table 2) [19]. Moreover, negative results were also obtained when human lymphoblastoid TK6 cells were exposed to different concentrations of TiO_2_NPs [20]. Nevertheless, both studies argued that the negative results were an artifact due to the interference of the TiO_2_ fluorescence with the cytometry equipment. Authors support their view based on the positive effects observed when they used the classical CBMN assay, although in this case cytochalasin B was used to arrest cytokinesis.

Regarding ZnONPs, although several studies used the CBMN assay to detect their potential genotoxicity, no studies used FC to detected MN induction. According to the reported data, the genotoxicity of ZnONPs can also be considered a controversial issue. In a wide study using two sizes (20 and 70 nm), both positively and negatively charged, no genotoxicity was observed *in vitro*, using the chromosomal aberration test [39]. Negative results were also obtained in exposed mice where no MN induction was observed [39]. These in vivo negative effects agree with the previous data reported after single oral ingestion [40]. Nevertheless, positive effects were found in Caco-2 cells [41] and in human THP1 monocytic cells [42] by using the CBMN assay. This genotoxic effect was attributed to their ability to induce reactive oxygen species (ROS) [41,42,43], generating an inflammatory response [42]. ZnONPs dissolve quickly in both culture medium and intracellularly and after 48 h more than 80% of the ZnONPs dissolve to their ionic form [43]. Nevertheless, neither of the reported studies proved the ability of Zn ions to produce MN. Hence, we propose that the ions release would explain the in general controversial data reported when the genotoxicity of ZnONPs is evaluated after acute exposures, the negative effects observed after long-term exposures [43] and the negative effects observed in in vivo studies. Furthermore, as it is well known, ZnONPs are highly cytotoxic which could mask a genotoxic response [44,45]. In this way, positive induction of MN was reported in BEAS-2B cells but only at the highest ZnONPs tested concentration and in the presence of bovine serum albumin in the culture medium [46]. Despite the fact that we previously demonstrated the uptake of ZnONPs by BEAS-2B cells [5], in the present work we did not get a significant induction of MN. Although in a non-significant manner, high values of MN formation were observed at the highest tested dose, which also produced toxicity. In this way, our results would confirm the previously indicated associating increased MN values with high toxicity [46]. It should be pointed out that BEAS-2B cells have an important antioxidant capacity in vitro [47] what would make these cells to show greater resistance.

Few studies on cerium dioxide nanoparticles have used micronucleus as a biomarker of genotoxicity and none of them used the FCMN assay. Although CeO_2_NPs are proposed as a potent antioxidant, some discrepant results exist because prooxidant effects have also been reported. This dual mechanism of action has been proposed to be associated with cell characteristics, with prooxidant effects emerging in tumoral cells. Nevertheless, a recent study using a wide set of cell lines and different culture conditions concludes that no prooxidant effects were observed, regardless of the conditions/cells used [48]. According to that, “*antigenotoxicity*” more than genotoxicity has been reported for CeO_2_NPs. In BEAS-2B cells antigenotoxic effects were reported but using the comet assay [49]. These protective effects were also detected using the CBMN assay in cells exposed to ionizing radiation such as osteoblastic cells [50] and in human lymphocytes [51]. Nevertheless, positive induction of MN was reported in human neuroblastoma cells [52] and in human dermal fibroblasts [53], as well as in Wistar rats [54]. However, our results indicate that CeO_2_NPs exposure is unable to induce significant increases in the frequency of MN induction, as detected by using the FCMN assay, even though we have seen CeO_2_NPs uptake by the BEAS-2B [5]. Moreover and in spite of the high uptake observed in undifferentiated Caco-2 cells by using confocal microscopy, CeO_2_NPs were also unable to induce DNA damage as detected by the comet assay [55]. This lack of genotoxicity was associated with decreased levels of oxidized DNA bases, confirming the antioxidant properties of CeO_2_NPs.

Silver nanoparticles have been largely studied from the genotoxic point of view, including their ability to induce MN, as detected using flow cytometry approaches [20,21,22,23]. In our study, AgNPs induced significant increases in the frequency of MN but without a clear direct concentration-effect relationship. According to this, a previous work detected intracellular particle localization when exposed BEAS-2 to 10 µg/mL of AgNPs [12]. Using the atomic absorption spectroscopy (AAS) analysis, the authors were able to detect an average of 10 pg/cell of 50 nm uncoated AgNPs. This would confirm our recent studies in Caco-2 cells where AgNPs induced DNA strand breaks in differentiated cells, mediated by oxidative DNA damage [56], increasing a wide set of cell-transformation biomarkers in long-term exposed undifferentiated cells [57]. The induction of MN by AgNPs has been attributed to size, with MN induction at the small size (20 nm) in both Caco-2 and HepG2 cells but only MN induction in HepG2 cells at a larger size (50 nm) [21,23]. Nevertheless, an opposite effect was reported using the CBMN assay, where the induction of MN increased with the size of the used AgNPs [58]. Although the effects of AgNPs have been attributed to their ability to release silver ions, as demonstrated by the positive induction of MN by silver nitrate [59], negative effects were reported when silver acetate was used as a source of silver ions [22]. In fact, the negative observed effects were attributed to the use of polyvinylpyrrolidone (PVP)-coated AgNPs, which would avoid ions release [60]. Nevertheless, other studies using such type of coated AgNPs have reported positive MN induction in the CBMN assay [20]. Interestingly, a recent study using 5 and 50 nm citrate coated AgNPs reported a non-induction of MN, as measured by the FCMN assay [61], which emphasizes silver ions release as the underlying mechanism of AgNPs genotoxicity. According to that, a Trojan-horse-effect is accepted to explain the genotoxic effects of AgNPs, where their uptake would be followed by a release of silver ions.

MWCNTs have been largely evaluated from the genotoxic point of view, including the use of the micronucleus assay; nevertheless, neither of the reported studies used flow cytometry to quantify MN induction. According to the expected exposure route, all the in vitro studies used lung cells such as A549 and BEAS-2B cells. In a study using both cell lines and two different MWCNT (NM401 and NM402) positive MN induction was detected in A549 cells when exposed to the selected MWCNTs; however, negative effects were observed in the BEAS-2B cells [62]. Our results show a positive MN induction when using the FCMN assay, although without a clear dose-response relationship. Nevertheless, our results do not agree with those reported in the same cell line using two different commercially available MWCNTs [63]. On the other hand, these results agree with previous data also reported in A549 cells [64,65], where authors indicated that mechanisms associated with MN induction, associated with MWCNTs exposure, probably involve oxidative stress and inflammatory responses. In addition, the involvement of aneugenic mechanisms was also proposed [64]. Interestingly, the effects of MWCNTs in Chinese hamster lung cells were characterized by the formation of polyploidy and an increased number of bi- and multi-nucleated cells but without micronuclei induction [66]. When the induction of MN was evaluated in mice, as in vivo model, negative effects were observed in blood cells and in bone marrow cells, indicating a lack of systemic effects. Nevertheless, positive MN induction was observed in broncho-alveolar cells, as a target site of the MWCNTs exposure [63]. These results differ from those reported in the bone marrow of exposed mice [67]. In that case, higher effects of functionalized MWCNTs than of non-functionalized ones were reported. Our positive results, although with a no clear dose-response relationship, would confirm the potential genotoxicity of MWCNT, their ability to induce MN and the advantages of the flow cytometry to detect such effects.

Although we have not carried out in parallel one study using the standard CBMN assay, to prove the relevance of the FCMN data over the classical CBMN assay, we are reporting a comparative table (Table 2) where our qualitative FCMN data are compared with those obtained in the frame of the UE project using the CBMN assay in BEAS-2B cells. Is must be emphasized that both studies used the same cell line and the same batches of the selected nanomaterials. In addition, the same cell culture protocol was also used. According to this data, the FCMN assay was able to detect the genotoxic potential of NM101 and NM401 that were unnoticed when the CBMN was used. On the other hand, the reported negative data obtained for NM110 with the FCMN assay were not observed in the CBMN assay, where slight but significant effects were observed. Since in general, the MN induction produced by the selected nanomaterials is weak, the observed differences can be attributed to the large sampling size used by the FCMN assay (20,000 cells) regarding the 1000 binucleated cells normally used in the CBMN assay.

As a summary of this study, we can conclude that the FCMN assay is a very good tool to evaluate the ability of NMs to produce chromosomal damage. In addition to its speed and a large number of scored cells, the obtained results basically agree with those reported in the literature using the CBMN in vitro assay or using in vivo mammalian model organisms. By using this method we have confirmed the ability of AgNPs and MWCNTs to induce MN, although with a lack of dose-response relationship and non-significant effects for TiO_2_NPs, ZnONPs and CeO_2_NPs exposures. Nevertheless, according to the induced increases in the MN frequency and their concentration-effect relationships, the genotoxicity of AgNPs and MWCNTs must be considered as weak.

## Figures and Tables

**Figure 1 nanomaterials-09-01677-f001:**
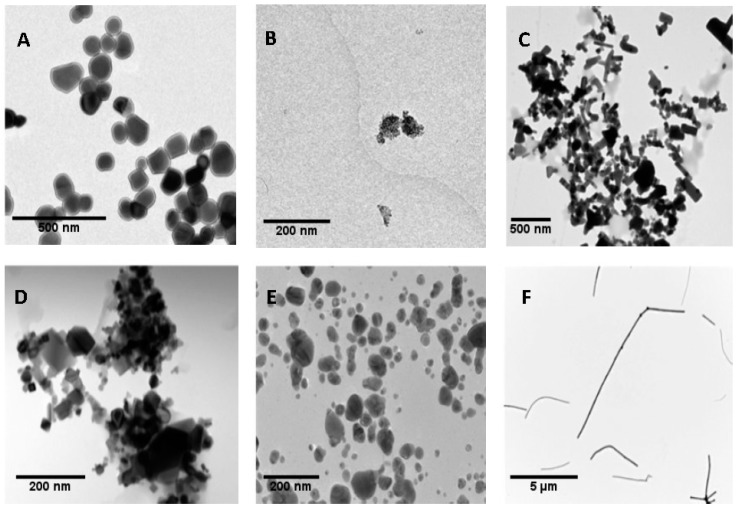
Transmission electron microscopy (TEM) images of dried nanoparticles (NPs). TiO_2_NPs NM100 (**A**), TiO_2_NPs NM101 (**B**), ZnONPs NM110 (**C**), CeO_2_NPs NM212 (**D**), AgNPs NM300K (**E**) and multi-walled carbon nanotube (MWCNT) NM401 (**F**).

**Figure 2 nanomaterials-09-01677-f002:**
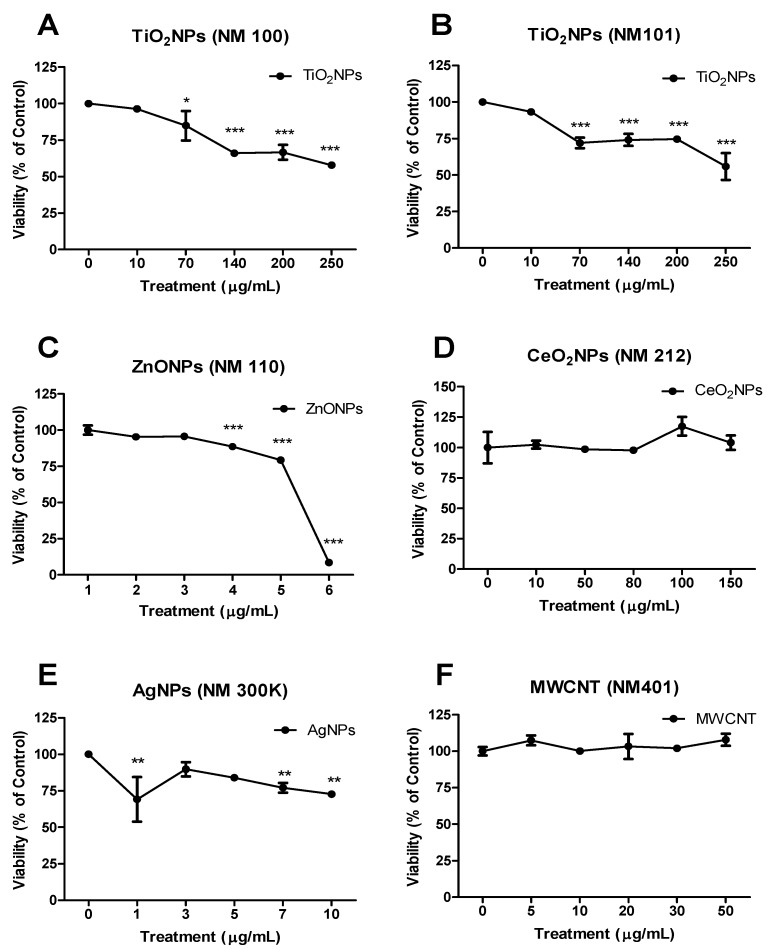
Viability curves tested in BEAS-2B. (**A**) TiO_2_NPs (NM100), (**B**) TiO_2_NPs (NM101), (**C**) ZnONPs (NM110), (**D**) CeO_2_NPs (NM212), (**E**) AgNPs (NM300K) and (**F**) MWCNTs (NM401). Results are plotted as mean ± SEM of two independent experiments. * *p* ≤ 0.05, ** *p* ≤ 0.01, *** *p* ≤ 0.001 (one way-ANOVA).

**Figure 3 nanomaterials-09-01677-f003:**
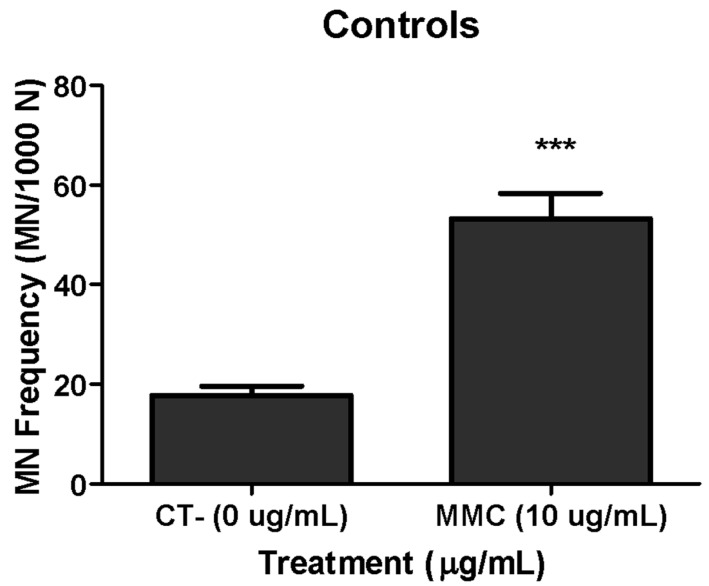
Validation of the assay testing the ability of the flow cytometry micronucleus (FCMN) assay, using MMC as a positive control, in BEAS-2B. Data of three independent experiments are represented as mean ± SEM. *** *p* ≤ 0.001 (*t*-test).

**Figure 4 nanomaterials-09-01677-f004:**
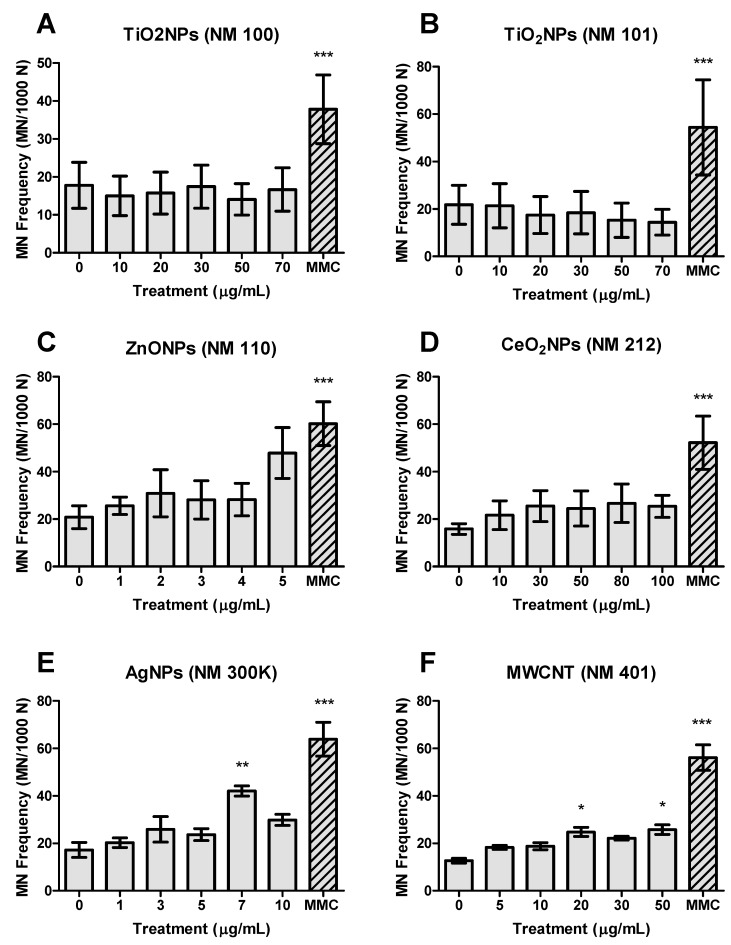
Micronucleus (MN) frequency (NM/1000 N) measured in BEAS-2B. (**A**) TiO_2_NPs (NM100), (**B**) TiO_2_NPs (NM101), (**C**) ZnONPs (NM110), (**D**) CeO_2_NPs (NM212), (**E**) AgNPs (NM300k) and (**F**) MWCNTs (NM401). Data of three independent experiments are represented as mean ± SEM. * *p* ≤ 0.05, ** *p* ≤ 0.01, *** *p* ≤ 0.001 (one way-ANOVA).

**Table 1 nanomaterials-09-01677-t001:** Nanoparticles characterization. NPs diameter analyzed by TEM in their dried forms and their hydrodynamic size, analyzed by dynamic light scattering (DLS). Data are represented as mean ± SEM.

NP	TiO_2_ NM100	TiO_2_ NM101	ZnO NM110	CeO_2_ NM212	Ag NM300K	MWCNT * NM401
TEM	104.01 ± 39.42	54.69 ± 35.39	132.37 ± 69.53	70.33 ± 49.61	7.75 ± 2.48	6012.09 ± 4091.45
DLS (10 μg/mL)	195.3 ± 2.50	152.2 ± 62.28	171.17 ± 13.61	172.2 ± 45.24	19.02 ± 1.78	-

* Average length.

**Table 2 nanomaterials-09-01677-t002:** Comparative data summarizing the obtained results (FCMN) with the overall results obtained in the frame of the UE Nanoreg project with BEAS-2B cells in the CBMN assay.

Nanomaterial	FCMN Assay	CBMN Assay
NM100	−	−
NM101	− (+ *)	−
NM110	−	+/−
NM212	−	nd
NM300K	+/−	−
NM401	+/−	−

−: negative; +: positive; +/−: weak positive; nd: no data; * results from D Bucchianico et al., 2017.

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
