# Peer review of "Micronuclei Detection by Flow Cytometry as a High-Throughput Approach for the Genotoxicity Testing of Nanomaterials"

_nanomaterials, 2019, doi:10.3390/nano9121677_

Round 1
Reviewer 1 Report
The paper is generally clear and well written, and the results suggest that a flow-cytometry based micronucleus assay may be a useful tool for relatively rapid screening of nanomaterials for potential genotoxicty.
A few comments/suggestions for improvement are listed below:
Preliminary cytotoxicity data were used to inform the concentrations used in the main test. It would be helpful to include criteria that were following in using these data to select the concentrations that were used for the main test (e.g. was the aim to produce no more than X% cytotoxicity?) It appears that dispersion of of the NPs was assessed prior to treatment but not after. It might be useful to include a comment on this. Cellular uptake is important in interpreting results but does not appear to have been assessed. It would be useful to discuss this more clearly. Line 243 states that high uptake in BEAS-2B cells was demonstrated with TiO2 NPs. It appears that this was in a previous study, but this could be made clearer. It may be helpful to more clearly indicate which NPs have uptake data, whether the data are with BEAS-2B cells or others, and to commment on the relevance (of lack thereof) of uptake data in other cell lines (comment is made about uptake in Caco 2 cells for some NPs). It would be useful to present the lower end of the confidence bars in Figure 4 to get a better picture of the variability of the data. I am confused by Table 2. This table states that positive results were seen in the FCMN with NM101 (and this is also stated on line 366-367), but this conflicts with the negative results shown in Figure 4B. If the table is referring to results from a separate study this should be made clear. Given the lack of dose-response with AgNPs and MWCNTs, I would argue that these results are equivocal rather than clear evidence of a positive genotoxic effect, as per OECD Test Guidelin 478. This should be discussed in the paper, rather than suggesting that the results are clearly positive with these substances.
Author Response
Manuscript ID: nanomaterials-630122
Type of manuscript: Article
Title: Micronuclei detection by flow cytometry as a high-throughput approach for genotoxicity testing of nanomaterials
Authors: Alba García Rodríguez, Liliya Kazantseva, Laura Vila, Laura Rubio, Antonia Velázquez, María José Ramírez de Haro, Ricard Marcos*, Alba Hernández*
RESPONSE TO REVIEWERS
REVIEWER 1.
Open Review
(x) I would not like to sign my review report
( ) I would like to sign my review report
English language and style
( ) Extensive editing of English language and style required
( ) Moderate English changes required
(x) English language and style are fine/minor spell check required
( ) I don't feel qualified to judge about the English language and style
|
Yes |
Can be improved |
Must be improved |
Not applicable |
|
|
Does the introduction provide sufficient background and include all relevant references? |
(x) |
( ) |
( ) |
( ) |
|
Is the research design appropriate? |
(x) |
( ) |
( ) |
( ) |
|
Are the methods adequately described? |
(x) |
( ) |
( ) |
( ) |
|
Are the results clearly presented? |
(x) |
( ) |
( ) |
( ) |
|
Are the conclusions supported by the results? |
( ) |
(x) |
( ) |
( ) |
Comments and Suggestions for Authors
The paper is generally clear and well written, and the results suggest that a flow-cytometry based micronucleus assay may be a useful tool for relatively rapid screening of nanomaterials for potential genotoxicity.
RESPONSE:
We thanks the consideration of this general comment of the reviewer
A few comments/suggestions for improvement are listed below:
Preliminary cytotoxicity data were used to inform the concentrations used in the main test. It would be helpful to include criteria that were following in using these data to select the concentrations that were used for the main test (e.g. was the aim to produce no more than X% cytotoxicity?)
RESPONSE:
We thanks the reviewer for this advice. The corresponding information is now provided in the section 2.4, as follows: The criteria followed to choose the concentrations was deducted from the study of Bryce et al. (2007), where treatments exhibiting mean relative survivals ≥ 40% were carried forward for further processing [27].
It appears that dispersion of the NPs was assessed prior to treatment but not after. It might be useful to include a comment on this.
RESPONSE:
Dispersion measurements after exposure are not usually carried out. To be able to perform such measurements, by using the DLS approach, samples would need to be resuspended, vortexed, etc, which can lead to artificial NPs dispersion data. Alternatively, to determine the stability of the dispersions, measurements over time can be carried out. We have do it in a previous study and this is now indicated in the section 3.1. as follows: To evaluate the effects of time on NPs dispersion, in a previous study DLS measurements were carried 24 h after dispersion. Results did not seem to indicate that the incubation time has a relevant effect on the NMs agglomeration [26].
Cellular uptake is important in interpreting results but does not appear to have been assessed. It would be useful to discuss this more clearly.
RESPONSE:
We already assessed this aspect in an earlier published paper [5] working with the same NMs; so, we did not think that was it necessary to repeat the experiments. However, we mentioned and cited this work in the 3rd paragraph of the Introduction section.
Line 243 states that high uptake in BEAS-2B cells was demonstrated with TiO2 NPs. It appears that this was in a previous study, but this could be made clearer. It may be helpful to more clearly indicate which NPs have uptake data, whether the data are with BEAS-2B cells or others, and to comment on the relevance (of lack thereof) of uptake data in other cell lines (comment is made about uptake in Caco 2 cells for some NPs).
RESPONSE:
As mentioned before, our previous results in the NPs uptakes have been mentioned either in the Introduction as well as in the Results and Discussion section for each evaluated NPs.
It would be useful to present the lower end of the confidence bars in Figure 4 to get a better picture of the variability of the data.
RESPONSE:
We thanks for the suggestion. The figure has been modified according to this.
I am confused by Table 2. This table states that positive results were seen in the FCMN with NM101 (and this is also stated on line 366-367), but this conflicts with the negative results shown in Figure 4B. If the table is referring to results from a separate study this should be made clear.
RESPONSE:
We thanks for this finding. Yes, this positive results are originally from another independent study. The error/confusion has been solved in both the table (by adding an asterisk (*) indicating the origin of the data), and in the text (mentioning that these results were also highlighted in Table 2).
Given the lack of dose-response with AgNPs and MWCNTs, I would argue that these results are equivocal
rather than clear evidence of a positive genotoxic effect, as per OECD Test Guideline 478. This should be discussed in the paper, rather than suggesting that the results are clearly positive with these substances.
RESPONSE:
It is true that there is no an evident concentration-dependent relationship for both AgNPs and MWCNTs compounds. Nevertheless, taking into account all the range of used concentrations, a trend to increase the effect when the concentration increases is observed, reaching statistical significance in one concentration (7 µg/mL) for AgNPs and two (20 and 50 µg/mL) for MWCNTs. Since the induced effects can be identified as a weak, the sampling error can produce both positive and negative results, moving close the borderline significance value. Nevertheless, in the summing up section we have modulated our classification of the genotoxic potential of both nanomaterials

Reviewer 2 Report
Comments:
This manuscript, ID nanomaterials-630122, titled “Micronuclei detection by flow cytometry as a high-throughput approach for genotoxicity testing of nanomaterials " for the journal nanomaterials. This study presents the use of the flow cytometric micronucleus (FCMN) assay to assess genotoxicity of various nanostructured materials using human bronchial epithelial cell line BEAS-2B, as a model system. This work should be fit for publication after the following significant revisions:
The overall quality of language needs to be improved, including revision of grammatical errors throughout the document. It would be informative to include somewhere in the introduction a statement defining what is new about this study. It would also be informative to include at the end of the introduction, the hypothesis this study is based on. Having two highly cited and highly similar abbreviations can be confusing. Specifically, MN and NM. In the introduction dispersion procedures are discussed, but it is not clear how the method presented here improves or even affect this technique. The EU project NANoREG is mentioned in several occasions, but its description or relevance to this work is not defined. It would be informative to mention the criteria behind choosing the BEAS-2B cell line as the model system. Figure 1: There is much variability in the magnification and scale bars of TEM images. Figure 2: The results presented here are based on only two experiments. I assume that these two experiments represent biological replicates. It is not clear if there were technical replicates as well (e.g., multiple well within the same plate). Still, it is worrisome to present P-values based on only two data points, and reach conclusions based on that. Figure 3: The meaning of the data represented, and error bars should be defined. The meaning of the asterisks should also be defined. It was done in previous figures but missed on this one. Results and Discussion: Genotoxicity – The structure of this section could be improved by first presenting the results obtained in this study first, followed by a discussion of the results and how these compare to published data. The discussion becomes confusing when it is not done this way. Page 10 – Line 234: The authors state that “a clear conclusion has not been reached until now.” This statement should be clearly supported by the facts. Page 12 – Line 288: It is stated that the cytotoxicity of ZnO NPs is “well-know”, without citing a reference. A proper reference should be added, supporting this statement. Also, the effect of high cytotoxicity masking high genotoxicity should also be supported with an appropriate reference. Results and Discussion: Genotoxicity – Throughout this section data from cancer cells line is presented as supporting or conflicting with the results reached in this study. The difference or similarity of the results should be discussed in light of the difference between BEAS-2B and these cancer cell lines.
Author Response
Manuscript ID: nanomaterials-630122
Type of manuscript: Article
Title: Micronuclei detection by flow cytometry as a high-throughput approach for genotoxicity testing of nanomaterials
Authors: Alba García Rodríguez, Liliya Kazantseva, Laura Vila, Laura Rubio, Antonia Velázquez, María José Ramírez de Haro, Ricard Marcos*, Alba Hernández*
RESPONSE TO REVIEWERS
REVIEWER 2.
Open Review
(x) I would not like to sign my review report
( ) I would like to sign my review report
English language and style
( ) Extensive editing of English language and style required
(x) Moderate English changes required
( ) English language and style are fine/minor spell check required
( ) I don't feel qualified to judge about the English language and style
|
Yes |
Can be improved |
Must be improved |
Not applicable |
|
|
Does the introduction provide sufficient background and include all relevant references? |
( ) |
(x) |
( ) |
( ) |
|
Is the research design appropriate? |
( ) |
(x) |
( ) |
( ) |
|
Are the methods adequately described? |
( ) |
(x) |
( ) |
( ) |
|
Are the results clearly presented? |
( ) |
( ) |
(x) |
( ) |
|
Are the conclusions supported by the results? |
(x) |
( ) |
( ) |
( ) |
Comments and Suggestions for Authors
Comments:
This manuscript, ID nanomaterials-630122, titled “Micronuclei detection by flow cytometry as a high-throughput approach for genotoxicity testing of nanomaterials" for the journal nanomaterials. This study presents the use of the flow cytometric micronucleus (FCMN) assay to assess genotoxicity of various nanostructured materials using human bronchial epithelial cell line BEAS-2B, as a model system. This work should be fit for publication after the following significant revisions:
The overall quality of language needs to be improved, including revision of grammatical errors throughout the document.
RESPONSE:
We thanks for this indication. All the manuscript has been carefully checked and those observed grammatical errors have been corrected
It would be informative to include somewhere in the introduction a statement defining what is new about this study. It would also be informative to include at the end of the introduction, the hypothesis this study is based on.
RESPONSE:
The proposed statement has been introduced at the end of the Introduction section.
Having two highly cited and highly similar abbreviations can be confusing. Specifically, MN and NM.
RESPONSE:
We thanks for this observation but we cannot do much more since in the rest of the literature both abbreviations are used this way
In the introduction dispersion procedures are discussed, but it is not clear how the method presented here improves or even affect this technique. The EU project NANoREG is mentioned in several occasions, but its description or relevance to this work is not defined. It would be informative to mention the criteria behind choosing the BEAS-2B cell line as the model system.
RESPONSE:
We thanks this types of comments. Briefly, the methodology for NPs dispersion presented here has been applied basically with the intention to harmonize intra and interlaboratory the way to treat and use NPs. This has been added and mentioned in the introduction section (at the end of the second paragraph).
The criteria behind choosing BEAS-2B is that since this model system has been previously tested in a vast number of nanotoxicological studies, there is a well background of literature to compare between methodologies and experiments. This has been mentioned in the 2.2, Cell culture sub-section in Materials and Methods section.
Figure 1: There is much variability in the magnification and scale bars of TEM images.
RESPONSE:
We thanks this comment. We would like also to provide consistency in the images format, but taking into account that every NPs behave distinctly from another ones due to its structural composition, we came up providing the best images we could take. We apologize for that.
Figure 2: The results presented here are based on only two experiments. I assume that these two experiments represent biological replicates. It is not clear if there were technical replicates as well (e.g., multiple well within the same plate). Still, it is worrisome to present P-values based on only two data points, and reach conclusions based on that.
RESPONSE:
We thanks and apologize for the confusion. These results are based on two independent experiments with three replicates per experiment, finally reaching and N of 6. This has been now indicated in the Statistical Analysis subsection of the Materials and Methods section.
Figure 3: The meaning of the data represented, and error bars should be defined. The meaning of the asterisks should also be defined. It was done in previous figures but missed on this one.
RESPONSE:
We thanks this observation. The information has been added in the legend corresponding to Figure 3.
Results and Discussion:
Genotoxicity – The structure of this section could be improved by first presenting the results obtained in this study first, followed by a discussion of the results and how these compare to published data. The discussion becomes confusing when it is not done this way.
RESPONSE:
We thanks for this comment. Now we first describe the general results on the first paragraph of this section.
Page 10 – Line 234: The authors state that “a clear conclusion has not been reached until now.” This statement should be clearly supported by the facts.
RESPONSE:
We thanks for this suggestion. We have modified the sentence to better explain the controversy between genotoxic assays.
Page 12 – Line 288: It is stated that the cytotoxicity of ZnO NPs is “well-know”, without citing a reference. A proper reference should be added, supporting this statement. Also, the effect of high cytotoxicity masking high genotoxicity should also be supported with an appropriate reference.
RESPONSE:
We thanks your suggestion. The citations are already added to the document.
Results and Discussion: Genotoxicity – Throughout this section data from cancer cells line is presented as supporting or conflicting with the results reached in this study. The difference or similarity of the results should be discussed in light of the difference between BEAS-2B and these cancer cell lines.
RESPONSE:
We thanks this point of view. However, it was not our intention to discuss about the background of our cell line, or the in vitro system used to identify the genotoxicity of the used NPs. Our main focus is to defence the high throughput potential of the proposed experimental design. Obviously the contrast with the results obtained using other cell line is discussed elsewhere. The only reference to the characteristics of the used BEAS-2B cells is that they have an important antioxidant capacity in vitro [Kinnula et al., 1994] what would make these cells to show greater resistance than other cell lines. In fact, in a recent paper [see ref 31]
where the effects of eight NMs were compared in two different human lung epithelial cell lines (A549 and BEAS-2B), BEAS-2B cells being less sensitive than A549 cells.

Round 2
Reviewer 1 Report
The revisions made to the paper in response to both peer reviewers' comments have improved the paper. There are a small number of editorial/typographic/grammatical errors that still need to be corrected however. I also have a few remaining comments:
Table 2 has been amended to indicate the positive response with NM101 in the FCMN assay was from a study by D. Bucchianico et al (2017). I think the table should also indicate that the response in the current study was negative. For NM300K and NM401 I would suggest the results are listed as +/- given their equivocal nature as previously commented (lack of clear concentration-response etc.).
The discussion on pp15-16 about MWCNT is conflicting. On the penultimate line of p15 there is a statement about 'the negative effects observed in the present work', but midway through p16 there is a statement 'our positive results would confirm the potential genotoxicity of MWCNT...'. These conflicting statements should be revised. As previously commented the lack of a clear concentration-response should be mentioned, which perhaps suggests the results are equivocal.
In the final paragraph on p16 there is a statement about 'the FCMN was able to detect the genotoxic potential of NM101...' but this does not make clear here that the positive response was seen in a different study and that negative results were seen in the current study. This should be pointed out.
Author Response
RESPONSE TO REVIEWERS
REVIEWER 1.
The revisions made to the paper in response to both peer reviewers' comments have improved the paper. There are a small number of editorial/typographic/grammatical errors that still need to be corrected however. I also have a few remaining comments:
Table 2 has been amended to indicate the positive response with NM101 in the FCMN assay was from a study by D. Bucchianico et al (2017). I think the table should also indicate that the response in the current study was negative. For NM300K and NM401 I would suggest the results are listed as +/- given their equivocal nature as previously commented (lack of clear concentration-response etc.).
RESPONSE:
We agree with the opinion of the reviewer and, consequently, we have modified Table 2 as suggested. In addition in the text, in the part corresponding to the discussion of NM401 and in the summing up, we have emphasized the lack of dose-response relationship observed for NM300K and NM401.
The discussion on pp15-16 about MWCNT is conflicting. On the penultimate line of p15 there is a statement about 'the negative effects observed in the present work', but midway through p16 there is a statement 'our positive results would confirm the potential genotoxicity of MWCNT...'. These conflicting statements should be revised. As previously commented the lack of a clear concentration-response should be mentioned, which perhaps suggests the results are equivocal.
In the final paragraph on p16 there is a statement about 'the FCMN was able to detect the genotoxic potential of NM101...' but this does not make clear here that the positive response was seen in a different study and that negative results were seen in the current study. This should be pointed out.
RESPONSE:
We thank the comment and, consequently, we have revised and corrected this part of the manuscript. Now we have eliminated all the indicated inconsistencies, rewriting these sections.
Reviewer 2 Report
The previous concern: "The EU project NANoREG is mentioned in several occasions, but its description or relevance to this work is not defined," was not addressed. The response to keep the different scales of the TEM images as they are applies perhaps to only one of the figures which has a 5 um scale. The rest are, 0.5 um, 200 nm, 0.2 um and 100 nm, which can be represented by a 200 nm scale.
Author Response
RESPONSE TO REVIEWERS
REVIEWER 2.
The previous concern: "The EU project NANoREG is mentioned in several occasions, but its description or relevance to this work is not defined," was not addressed.
RESPONSE:
We are sorry for the mistake. Now, we have introduced a new paragraph at the end of the Introduction section that read as follows:
“To identify what environmental health and safety aspects of nanomaterials are relevant from a regulatory point of view, the NANoREG EU project was launched. One of the aims was to determine the suitability of different in vitro assays in terms of reliability and predictive value. In such context we assessed ….”
The response to keep the different scales of the TEM images as they are applies perhaps to only one of the figures which has a 5 um scale. The rest are, 0.5 um, 200 nm, 0.2 um and 100 nm, which can be represented by a 200 nm scale.
RESPONSE:
We have modified the scales in the new Figure 1.